# Did the COVID-19 Pandemic Increase the Incidence of Acute Macular Neuroretinopathy?

**DOI:** 10.3390/jcm10215038

**Published:** 2021-10-28

**Authors:** Georges Azar, Sophie Bonnin, Vivien Vasseur, Céline Faure, Flore Salviat, Catherine Vignal Clermont, Cherif Titah, Selim Farès, Elise Boulanger, Sabine Derrien, Aude Couturier, Amélie Duvilliers, Anthony Manassero, Rabih Hage, Ramin Tadayoni, Francine Behar-Cohen, Martine Mauget-Faÿsse

**Affiliations:** 1Clinical Investigative Platform Department, Adolphe de Rothschild Foundation, 75019 Paris, France; sbonnin@for.paris (S.B.); vvasseur@for.paris (V.V.); fsalviat@for.paris (F.S.); cvignal@for.paris (C.V.C.); ctitah@for.paris (C.T.); sfares@for.paris (S.F.); eboulanger@for.paris (E.B.); sderrien@for.paris (S.D.); acouturier@for.paris (A.C.); aduvilliers@for.paris (A.D.); amanassero@for.paris (A.M.); rhage@for.paris (R.H.); rtadayoni@for.paris (R.T.); mgfaysse@me.com (M.M.-F.); 2Anterior Segment Department, Adolphe de Rothschild Foundation, 75019 Paris, France; 3Ramsay Générale de Santé, Private Hospital Saint Martin, 14000 Caen, France; celinefaureb@yahoo.fr; 4Ophthalmology Department, Lariboisière Hospital—Assistance Publique-Hôpitaux de Paris, AP-HP, 75010 Paris, France; 5Ophthalmology Department, OphtalmoPôle, Hôpital Cochin, Assistance Publique-Hôpitaux de Paris, AP-HP, Université de Paris, 75014 Paris, France; francine.behar@gmail.com; 6Centre de Recherche des Cordeliers, Team 17, INSERM U1138, Université de Paris, 75006 Paris, France

**Keywords:** acute macular neuroretinopathy, AMN, COVID-19, SARS-CoV-2, retina, OCT angiography, pandemic

## Abstract

Background: Acute macular neuroretinopathy (AMN) is an increasingly diagnosed disorder associated with several diseases. The aim of this study was to report the incidence of AMN cases diagnosed during the 2020 coronavirus disease 2019 (COVID-19) pandemic year in a French hospital, and to describe their different forms. Methods: All patients diagnosed between 2019 and 2020, in Paris Rothschild Foundation Hospital, with AMN, paracentral acute middle maculopathy (PAMM) and multiple evanescent white dot syndrome (MEWDS) were retrospectively collected using the software Ophtalmoquery^®^ (Corilus, V1.86.0018, 9050 Gand, Belgium). Systemic and ophthalmological data from AMN patients were analyzed. Results: Eleven patients were diagnosed with AMN in 2020 vs. only one patient reported in 2019. The incidence of AMN significantly increased from 0.66/100,000 visits in 2019 to 8.97/100,000 visits in 2020 (*p* = 0.001), whereas the incidence of PAMM and MEWDS remained unchanged. Four (36%) of these AMN patients were tested for COVID-19 and received positive polymerase chain reaction (PCR) tests. Conclusions: The incidence of AMN cases increased significantly in our institution in 2020, which was the year of the COVID-19 pandemic. All AMN-tested patients received a positive COVID PCR test, suggesting a possible causative link. According to the different clinical presentations, AMN may reflect different severe acute respiratory syndrome coronavirus-2 (SARS-CoV-2) pathogenic mechanisms.

## 1. Introduction

In December 2019, after cases of serious illness causing pneumonia and death were first reported in Wuhan, China, the World Health Organization announced “coronavirus disease 2019 (COVID-19)” as a novel severe acute respiratory syndrome coronavirus (SARS-CoV), named “SARS-CoV-2” by the International Committee on Taxonomy of Viruses (ICTV) [1]. Apart from the sometimes fatal respiratory syndrome, COVID-19 leads to multi-visceral complications associated with inappropriate immune and inflammatory response [2], endothelial dysfunction and thromboembolic disorders [3]. Damage to the central and peripheral nervous system [4,5] was reported, including the autonomous nervous system that showed dysregulation at the early [6] and late phases of the disease [7,8].

Although arterial and venous thrombotic events have been reported, no clear increase in the incidence of retinal arterial or vein occlusion has been associated with COVID-19 infection. However, several cases of retinal disorders with evolving treatment paradigms occurring at the time of COVID-19 infection have been described [9,10,11]. A recent review confirmed that the majority of the retinal findings in COVID-19 patients were cotton wool spots, flame-shaped intraretinal hemorrhages, PAMM, AMN, or retinal vein occlusions. Rarely, inflammation involving the retina or the choroid, or reactivation of previously quiescent uveitis, may also be seen [12,13,14,15,16,17,18].

The exact incidence of these retinal disorders during the COVID-19 pandemic, commonly associated with viral infections or related to vascular dynamic changes, such as paracentral acute middle maculopathy (PAMM), multiple evanescent white dot syndrome (MEWDS) and acute macular neuroretinopathy (AMN), has not been yet reported.

PAMM is characterized by parafoveal hyper-reflective band-like lesions visible at the level of the inner nuclear layer (INL) on spectral-domain optical coherence tomography (SD-OCT) in patients with acute negative scotoma. The lesion appears grey with near-infrared reflectance imaging. A vascular origin has been hypothesized [19].

MEWDS mostly affects healthy middle-aged females who present with photopsia, dyschromatopsia, or scotoma, most frequently unilaterally, following a viral episode. On fundus exam, grey-white lesions located in the posterior pole and eventually associated with moderate inflammatory signs are seen. On SD-OCT, discontinuities of the ellipsoid zone (EZ) and of the external limiting membrane (ELM) are transient and regress with possible thinning of the outer nuclear layer (ONL) [20].

AMN is a rare disorder of the outer retinal layers, diagnosed by multimodal imaging. Fundus photos usually show dark red petaloid lesions around the fovea, and the diagnosis is confirmed by: (1) infrared scanning laser ophthalmoscopy (SLO) that shows hyporeflective parafoveal wedge-shaped or teardrop-shaped areas [21,22] and (2) SD-OCT B-scans that show a hyper-reflective lesion at the level of the outer plexiform layer (OPL)-ONL, which corresponds to the lesions seen on infrared SLO images. The EZ, often disrupted at the acute phase, usually slowly recovers during the evolution of the disease, whereas the ONL usually becomes thinner [21,23]. Although the exact mechanism of AMN is unknown, an acute suffering of the outer retina secondary to the circulatory deregulation of the deep capillary plexus has been hypothesized [24,25], but a deregulation of blood flow in the choroid, which supplies oxygen and nutrients to the photoreceptors, could also be advocated. Indeed, AMN has been associated with vascular factors (hypo- or hypertension, sympathomimetic drugs use, anaphylaxis, thrombocytopenia, anemia, hyperviscosity, hypovolemia, and dehydration) [24,25] but also with optic neuritis [26], as well as with various infectious agents such as dengue fever [27], influenza virus [28], and with vaccination [29,30].

During 2020, when the SARS-CoV-2 pandemic was high in France, we observed a significant increase in AMN cases in Rothschild Foundation Hospital. The purpose of this work was to describe and compare the incidence of AMN, PAMM, and MEWDS between 2020 and the previous year, to describe the different forms of AMN encountered, and finally to provide a potential mechanism that might link AMN to SARS-CoV-2 infection.

## 2. Materials and Methods

### 2.1. Study Design and Patient Selection

This was a retrospective observational noncomparative monocentric study. Patients diagnosed with AMN, PAMM, and MEWDS, who presented to Rothschild Foundation Hospital, Paris, France, between 2019 and 2020, were retrospectively evaluated. The medical charts of eligible patients were searched for using search terms and codes through the Ophtalmoquery^®^ electronic theater software (Corilus, V1.86.0018, 9050 Gand, Belgium). Approval was obtained by the Rothschild Foundation Hospital review board—IRB 00012801—under the study number CE_20210126_3_MMT. The study adhered to the tenets of the Declaration of Helsinki, as well as the International Conference on Harmonization Good Clinical Practice guidelines.

### 2.2. Data Collection

Demographic data from AMN cases were collected including sex, age, and the presence of systemic comorbidities such as systemic hypertension and diabetes mellitus. Data concerning ophthalmological evaluation at baseline and during follow-up were also collected. This included measurement of best-corrected visual acuity (BCVA), thorough slit-lamp examination, intraocular pressure (IOP) measurement using Goldmann applanation, and direct fundus ophthalmoscopy. Moreover, procedure analysis from fundus color photography, infrared SLO, SD-OCT B-scan, fundus auto fluorescence (FAF), OCT-angiography (OCTA), fundus fluorescein angiography (FFA), and indocyanine green angiography (ICGA) (Heidelberg Engineering, V1.10.12.0, 69115 Heidelberg, Germany) were also recorded. Patients diagnosed with AMN presented with near infrared parafoveal hypo-reflective lesions, hyper-reflectivity of the OPL, and attenuation of the ellipsoid and interdigitation zones on SD-OCT B-scans. Those AMN cases were classified into two groups (Figure 1): 1—the “classical” AMN group (Figure 1A) who presented with a petaloid, sharp border lesion on infrared SLO (that corresponded to the hyper-reflectivity of the OPL on SD-OCT B-scan), and always with disruption of the extrafoveal EZ; and 2—the AMN “plus”, which was further categorized into 2 sub-groups: (i) AMN “plus” with “photoreceptoritis” aspect (Figure 1B) defined by a petaloid, fluffy border lesion on infrared SLO, OPL hyper-reflectivity on SD-OCT B-scan, and EZ disruption observed in the foveal center, and (ii) AMN “plus” with “photoreceptoritis” and “retinal pigment (RP) epithelitis” (Figure 1C), which involves retinal pigment epithelium (RPE). Demographic data of control patients who met the clinical criteria of PAMM and PEWDS were also collected. All control patients received the same ophthalmological evaluation that was conducted with AMN patients at baseline and during follow-up.

### 2.3. Statistical Analysis

Statistical analysis was performed using IBM SPSS 22.0 (SPSS Inc., Chicago, IL, USA), and graphics were developed using GraphPad Prism 6. The results were expressed as means ± standard deviation (SD), and percentages with confidence intervals of 95%. Qualitative data were compared using Chi-square and Fisher’s exact test, and quantitative data using independent-samples *t* test and paired-samples t test. Differences with *p* values < 0.05 were considered statistically significant.

## 3. Results

Eleven patients developed AMN during the SARS-CoV-2 pandemic in 2020, compared to only one patient in 2019. Among those 11 AMN cases, there were 10 females (91%) and only one male (9%). The mean age of patients was 26 years (range: 17–38 years). The incidence rate of AMN significantly increased from 0.66/100,000 visits in 2019 to 8.97/100,000 visits in 2020 (*p* = 0.001). Table 1 compares the incidence variation between AMN, MEWDS, and PAMM in 2019 and 2020. Whilst AMN increased significantly, no significant difference was observed regarding the incidence of MEWDS and PAMM between 2019 and 2020 (3.95/100,000 visits in 2019 and 3.26/100,000 in 2020 for MEWDS (*p* > 0.99) vs. 3.29/100,000 visits in 2019 and 5.71/100,000 visits in 2020 for PAMM (*p* = 0.34)). Thus, AMN was the only outer retinal disease for which the incidence increased during the SARS-CoV-2 pandemic in our center.

All AMN patients complained of scotomas, which were paracentral in eight cases and central in all others (patients 2, 8, and 10). Six patients presented with “classical” AMN (patients 1, 4, 6, 8, 9, and 11), three patients had AMN “plus” with photoreceptor alterations (patients 2, 3, and 5), and two patients had AMN “plus” with photoreceptors and RPE alterations (patients 7 and 10). No patient presented with systemic high blood pressure, and only one patient had diabetes mellitus. The baseline characteristics of all patients are summarized in Table 2.

Among the four patients (36% of the total) who presented with positive polymerase chain reaction (PCR) tests (patients 2, 4, 9 and 11), three had “classic” AMN (patients 4, 9 and 11), and one (patient 2) had AMN “plus” with photoreceptors alterations. AMN was always concomitant with the extra-ocular signs of COVID-19 in these four patients, two of whom (patients 4 and 11) were also taking oral estrogen/progesterone. Of the seven other patients, who had not undergone PCR testing, five were asymptomatic (patients 5, 6, 7, 8, and 10) while two (patients 1 and 3) had signs in favor of flu or COVID-19. Two (patients 1 and 8) of these patients developed AMN during an episode of anti-MOG (myelin oligodendrocyte glycoprotein) acute optic neuritis. Moreover, one of these patients (patient 1) had meningitis 15 days before the onset of AMN. One patient (patient 10) had inactive proliferative diabetic retinopathy (PDR) treated previously with laser panretinal photocoagulation. One patient (patient 7) underwent a vaccination booster (Boostrix, against diphtheria, tetanus, and whooping cough) 8 weeks earlier and presented symptoms of migraine, facial palsy, and neurologic right finger paresthesia concomitant with the acute drop in visual acuity (left eye (OS) > right eye (OD)). This patient presented with PAMM in the right eye and AMN “plus” with RPE alterations in the left eye. Another patient presented with nystagmus and amblyopia (patient 2), explaining the poor BCVA in the right eye.

The color fundus photography, when available, was normal or showed discrete dark red petaloid lesions, while the infrared SLO images always showed typical petaloid hyporeflective lesions with sharp borders in the case of “classical” AMN and less distinct borders in the case of AMN “plus” cases. The FAF images were normal. The OCT B-scans in all cases showed a hyper-reflective lesion at the level of the OPL/ONL at the site of the hyporeflective lesions seen on the infrared SLO images. The EZ in the central area was normal in all the “classical” AMN cases, but was disrupted in all the AMN “plus” cases.

OCTA was performed in eight patients (all except patients 6, 8, and 10), all of whom showed a slight attenuation of the signal in the deep capillary plexus (DCP) in an area smaller than the area of hyporeflectivity seen on the infrared SLO images. Both the superficial and intermediate capillary plexuses were normal in all these cases (Figure 2). In addition, some areas of slight hyporeflectivity in the choriocapillaris were seen on the OCTA scans.

Follow-up data were only available for seven study patients (1, 2, 3, 4, 6, 8, and 11); mean follow-up in these patients was 128 days (range, 35-259 days). An improvement of the EZ with thinning of the corresponding ONL was observed in three “classical” AMN patients (1, 6, and 8), who all had a follow-up of more than 4 months. In cases 2 and 3 (AMN “plus”), the photoreceptor lesions disappeared with an ad integrum restoration of the macular zone. Patients 5, 7, 9, and 10 had no follow-up data.

## 4. Discussion

Among extrapulmonary manifestations caused by SARS-CoV-2, ocular manifestations, including retinal involvement, have been reported. Several reports showed cases with retinal disorders, mostly vaso-occlusive events, such as flame-shaped retinal hemorrhages, cotton wool spots, and retinal sectorial pallor areas [24,25,26,27]. One case of AMN [12] and three other cases with PAMM and AMN associated with COVID-19 have been reported [13,14].

As previously described, AMN may occur following dengue fever [25,27,29], influenza viral infection [28], and post-influenza vaccination [29,30,31], but the exact mechanism of this association is still unknown. Similarly, MEWDS patients may also present with a history of viral infection. In fact, some cases associated with a previous infection with a neurotropic Herpesviridae family virus have been described [32]. Interestingly, compared to AMN, the incidence of MEWDS (which is an RP epithelitis [33] supposed to be associated with antigen-driven inflammatory reaction and immunity disorders) and of PAMM (the mechanism of which is believed to be ischemic, involving the hypoperfusion of DCP [34]) did not increase during the pandemic in our center. This increase in incidence, attributed almost exclusively to AMN, would suggest specific mechanisms that might be associated with COVID-19 infection pathogenesis.

SARS-CoV-2 is known to enter host cells through the interaction of its spike protein with the entry receptor angiotensin-converting enzyme (ACE)-2 in the presence of transmembrane serine protease (TMPRSS) 2. Proposed mechanisms for COVID-19 caused by infection with SARS-CoV-2 include: 1—direct virus-mediated cell damage; 2—dysregulation of the renin–angiotensin–aldosterone system (RAAS) as a consequence of the downregulation of ACE-2 related to viral entry, which leads to decreased cleavage of angiotensin I and angiotensin II; 3—endothelial cell damage and thromboinflammation; and 4—dysregulation of the immune response and hyperinflammation caused by inhibition of interferon signaling by the virus, T-cell lymphodepletion, and the production of proinflammatory cytokines, particularly interleukin (IL)-6 and tumor necrosis factor (TNF)-α [2].

Interestingly, in the eye, ACE-2 receptors are expressed in the retinal ganglion cell layer, inner plexiform layer, inner nuclear layer, and photoreceptor outer segments. On the other hand, TMPRSS2 is also expressed in multiple retinal neuronal cells, vascular and perivascular cells, as well as in retinal Müller glial cells [35]. ACE-2 receptor expression was also shown in astrocytes and oligodendrocytes in human and rodent brains [36] and in microglial cells [37], suggesting possible viral entry in the retinal cells. Indeed, SARS-CoV-2 RNA was found in the retina of some patients who died from COVID-19 [38].

In addition to those gene expressions in the retina, COVID-19 has been shown to induce deregulation of the autonomous system that controls choroidal blood flow. This is in line with the observation that two of our patients presented an AMN during an episode of anti-MOG acute optic neuritis, since some of the choroidal nerves are known to be myelinated [39]. Therefore, alteration of choroidal nerve myelinization could induce both optic neuritis and choroidal perfusion changes. Deschamps et al. suggested that it was imperative to check for associated AMN in cases of acute optic neuritis, especially those associated with MOG antibodies [26]. Moreover, in 2020, Zhou et al. published the first case of a patient with SARS-CoV-2 infection and MOG-IgG antibody-mediated acute optic neuritis [40]. Since oxygen and nutrient supplies depend on choroidal blood flow, it is expected that any change in choroidal blood flow regulation could cause suffering of the outer retina, particularly hyperoxygenation, which would expose the retina to oxidative damage.

The other possible mechanism of AMN in the context of COVID-19 could be the hyperinflammatory response induced by SARS-CoV-2, which involves many inflammatory cells. Interestingly, resident microglia, known as innate immune cells, are found histologically in the eye around the retinal vessels and in the choroid. A dysregulated macrophage/microglia response can be damaging to the host, as it is seen in the macrophage activation syndrome induced by severe infections, including infections with the related virus SARS-CoV-2. Hyper-reflectivity of the OPL may then be an indirect sign of this activation associated with damage to adjacent tissues, i.e., the photoreceptors and, more severely, the RPE [41]. Moreover, photoreceptors and RPE inflammatory lesions, which were a subgroup of AMN seen in our series, may result from a more intense macrophage/microglial activation within the AMN syndrome initiated by direct virus-mediated cell damage of the outer retina, with endothelial cell damage and thrombo-inflammation in the terminal retinal deep capillary plexus. Importantly, microglia and retinal glial Müller (RMG) cell interactions could also explain the location of AMN lesions in the area of Z-shaped RMG cells [42]. Therefore, we speculate that this significant increase in AMN could be due to an autonomous system deregulation of choroidal blood flow caused by SARS-CoV-2 and/or inappropriate activation of the resident microglia [43].

Finally, we believe that thromboembolic events may also play a major role in SARS-CoV-2-induced AMN, which potentially explains the retinal findings seen in this particular setting. In fact, similarly to Zhang et al., who showed that COVID-19 could be responsible for coagulation disorders with an increase in the D-dimer levels and fibrinogen degradation products, leading thus to distal ischemia [44], we strongly believe that endothelial damage induced by the virus could be the key mechanism in the pathogenesis of COVID-19 in our classical AMN form, which could correspond to microangiopathic damage at the level of the deep capillary plexus. This conclusion is similar to the one discussed by Colmenero et al. in their evaluation of lesions seen in chilblains. They found that those lesions, commonly seen during the SARS-CoV-2 pandemic, correspond histopathologically to acro-ischemic lesions, and are due to the presence of viral particles in the endothelium, with evidence of vascular damage [45]. Those findings were reflected in our cases by the presence of hyporeflected areas in the choriocapillaris that were seen on OCTA scans.

Our study has several limitations. First, it includes a relatively small sample of patients, which is due to the low general incidence of AMN. Second, its retrospective and monocentric design does not necessarily permit the extrapolation of the data to the general population. Third, the duration of data collection was short, which is linked to the relatively recent start of the pandemic in 2019. Finally, seven of our patients could not be tested for COVID-19, which was partially due to the lack of tests at the time of presentation, and this might have systematically biased our results. Therefore, further continuous investigations with larger double-masked multicentric prospective studies remain crucial to fully validate our results.

In conclusion, we observed a significant increase in the incidence of AMN in our hospital during the COVID-19 pandemic. The exact mechanism by which the virus could cause AMN remains uncertain. Different pathogenetic hypotheses, including coagulation anomalies, hyperinflammation, and immune dysregulation, are proposed. Multicentric studies on a large scale are needed to ascertain the link between COVID-19 and AMN.

## Figures and Tables

**Figure 1 jcm-10-05038-f001:**
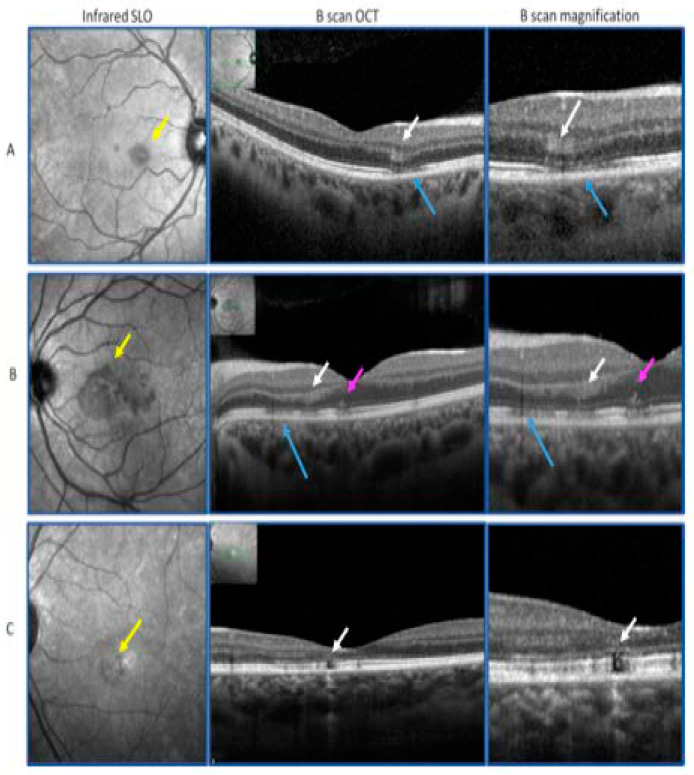
(**A**): “Classical” acute macular neuroretinopathy (AMN) with a petaloid, sharp border, lesion on infrared scanning laser ophthalmoscopy (SLO) (yellow arrow), corresponding to an outer plexiform layer (OPL) hyper-reflectivity on optical coherence tomography (OCT) (Heidelberg Engineering, V1.10.12.0, 69115 Heidelberg, Germany) B-scan (white arrows), and with extrafoveal ellipsoid zone (EZ) disruption (blue arrows). (**B**): AMN “plus” with “photoreceptoritis” with a petaloid, fluffy border, lesion on infrared SLO (yellow arrow) corresponding to OCT B-scan OPL hyper-reflectivity (white arrows), and EZ disruption observed in the foveal center (pink arrows). (**C**): AMN “plus” with “photoreceptoritis” and “retinal pigment epithelitis”: infrared SLO lesion (yellow arrow) with retinal pigment epithelium (RPE) atrophy on OCT B-scan (white arrows).

**Figure 2 jcm-10-05038-f002:**
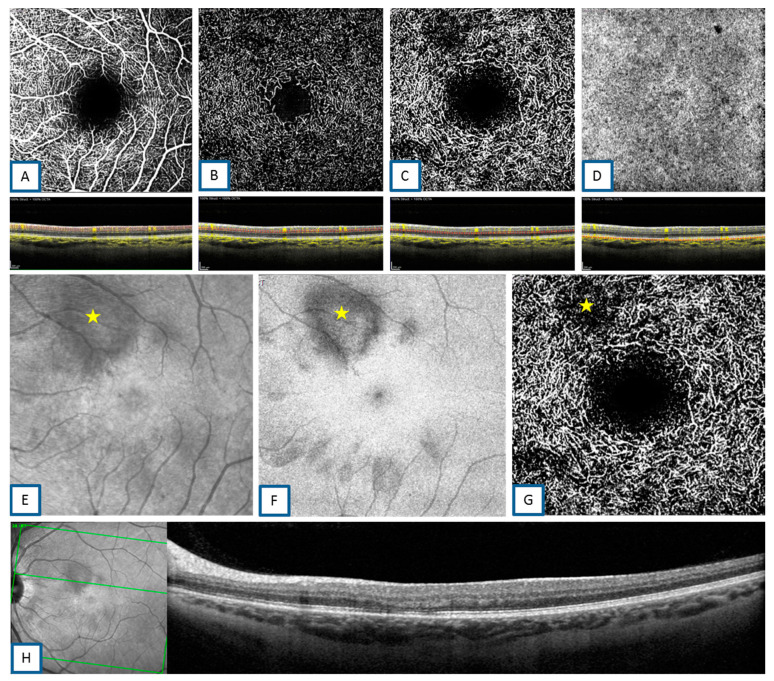
(**A**): Optical coherence tomography angiography (OCTA) (Heidelberg Engineering, V1.10.12.0, 69115 Heidelberg, Germany) superficial capillary plexus “en-face” and B-scans showing normal flow. (**B**): OCTA intermediate capillary plexus “en-face” and B-scans showing normal flow. (**C**): OCTA deep capillary plexus “en-face” and B-scans showing areas of reduced flow in the areas of infrared scanning laser ophthalmoscopy (SLO) hyporeflectivity. (**D**): OCTA choriocapillary “en-face” and B-scans showing normal flow. (**E**–**G**): AMN infrared SLO hyporeflectivity, ellipsoid lesion, and area of reduced flow, respectively, detected on infrared SLO, “en-face” spectral domain (SD)-OCT, and deep capillary plexus OCTA (yellow stars). The area of deep capillary flow reduction on OCTA is smaller than the area of hyporeflectivity seen on infrared SLO images. (**H**): SD-OCT B-scan of the corresponding AMN lesion.

**Table 1 jcm-10-05038-t001:** Prevalence of acute macular neuroretinopathy (AMN), paracentral acute middle maculopathy (PAMM) and multiple evanescent white dot syndrome (MEWDS) in 2019 and 2020 using the software Ophtalmoquery^®^ (Corilus, V1.86.0018, 9050 Gand, Belgium).

	Prevalence (for 100,000 Visits)		
	2019	2020	*p*
MEWDS	3.95	3.26	>0.99
PAMM	3.29	5.71	0.34
AMN	0.66	8.97	0.001

**Table 2 jcm-10-05038-t002:** Baseline characteristics of patients.

Number	Age (Years)	Sex	Laterality	COVID PCR Test	Observation	RE VA	LE VA	Clinical Form	Follow-Up (Days)
1	17	F	Bilateral	No test	MOG +	20/20	20/20	Classic	182
2	21	M	RE	Positive		20/2000	20/20	Photoreceptoritis	70
3	18	F	Bilateral	No test		20/20	20/20	Photoreceptoritis	259
4	28	F	Bilateral	Positive		20/20	20/20	Classic	35
5	38	F	LE	No test		20/20	20/32	Photoreceptoritis	LTFU
6	30	F	RE	No test		20/20	20/20	Classic	131
7	31	F	LE	No test	Vaccination	20/20	20/40	Epithelitis	LTFU
8	15	F	RE	No test	MOG +	20/2000	20/2000	Classic	159
9	27	F	Bilateral	Positive		20/16	20/16	Classic	LTFU
10	38	F	RE	No test	Diabetic retinopathy	20/2000	20/20	Epithelitis	LTFU
11	22	F	RE	Positive		20/20	20/20	Classic	60

Abbreviations: COVID: coronavirus disease; PCR: polymerase chain reaction; RE: right eye; VA: visual acuity; LE: left eye; F: female; M: male; MOG +: presence of anti-myelin oligodendrocyte glycoprotein antibodies; LTFU: lost to follow-up.

## Data Availability

The data in this study are available on request from the corresponding author.

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
