# Peer review of "Did the COVID-19 Pandemic Increase the Incidence of Acute Macular Neuroretinopathy?"

_jcm, 2021, doi:10.3390/jcm10215038_

Round 1

Reviewer 1 Report

The paper is interesting and well written, no comment for the Authors.

The topic is original and has never been addressed so far. The manuscript adds a specific putative mechanism of action of COVID as regards to the pathophysiology of AMN. The paper well written and the text clear and easy to read.  

Reviewer 2 Report

The authors hypothesized an increased incidence in acute macular neuroretinopathy (AMN) associated to the corona pandemic. To this purpose they analyzed retrospectively the prevalence of AMN, paracentral acute middle maculopathy (PAMM) and the multiple evanescent white dots syndrome (MEWDS) in one French hospital. They described three different observed forms of AMN and discussed potential pathogenic mechanisms linked to a SARS-CoV-2 infection and conclude to have seen a SARS-CoV2-induced AMN. Generally, I support the analysis of potential ocular manifestations of SARS-CoV”, the paper is well written and discusses nicely different potential pathogenic pathways. However, the small sample size collected in only 2 years from one hospital does not support a statistical analysis and the conclusion made by the authors.

Minor comments

On page 5, line 6 it is written “…follow up of ≥ to 131 days.” It seems there is a number missing.

Major comments

Introduction

  1. There is not massive but some literature about the potential ocular manifestations of SARS-CoV2 and these are not sufficiently presented. Please modify the introduction accordingly

Methods and materials

  1. Due to the fact that only 1 year with and 1 year without pandemic has been analyzed including in total 12 patients suffering from AMN, a statistical analysis seems not useful. A qualitative description of the results (except descriptive statistics like the mean age and similar) would be more useful. On the other hand, it might be possible to increase sample size by analyzing at least 2020 and 2021 as pandemic years and also 2 years before the pandemic.
  2. How many patients were analyzed suffering from PAM or MEWDS? Please add the missing numbers.

Results

  1. Demographics are not shown, though they might have a significant influence on the seen AMN incidence. Please add a table that shows the demographics comparing the three patient groups and comparing the assumed SARS-CoV2-negative and SARS-CoV2-positive patients. Especially the comorbidities like diabetes can have an impact on the AMN incidence, which would then not be increased due to a pathogenic mechanism in SARS-CoV2 but e.g., due to an increased number of SARS-CoV2 infected diabetic patients! Similar comorbidities in the other maculopathies would, on the other hand, support the hypothesis of the authors.

Discussion

  1. Regarding the small sample size (12 AMN patients), collected in only 2 years, in only 1 hospital, without confirmed exclusion of SARS-CoV2 in 7 patients (from the 11 AMN patients or all? Please clarify this), and incomplete demographic analysis, the data do not allow to conclude teh existence of a “SARS-CoV2-induced AMN”. The potential pathogenic mechanisms discussed are possible as are potential ocular manifestations in SARS-CoV2, but shown data only allow assumptions or hypothetic considerations. Thus, please rewrite the discussion coming to a more modest and conservative conclusion.
